# Astrocytic ApoE underlies maturation of hippocampal neurons and cognitive recovery after traumatic brain injury in mice

Tzong-Shiue Yu [1✉], Yacine Tensaouti[1], Elizabeth P. Stephanz[1], Sana Chintamen [1], Elizabeth E. Rafikian[2], Mu Yang[2] & Steven G. Kernie [1,3✉]

Polymorphisms in the apolipoprotein E (ApoE) gene confer a major genetic risk for the development of late-onset Alzheimer's disease (AD) and are predictive of outcome following traumatic brain injury (TBI). Alterations in adult hippocampal neurogenesis have long been associated with both the development of AD and recovery following TBI and ApoE is known to play a role in this process. In order to determine how ApoE might influence hippocampal injury-induced neurogenesis, we generated a conditional knockout system whereby functional ApoE from astrocytes was ablated prior to injury. While successfully ablating ApoE just prior to TBI in mice, we observed an attenuation in the development of the spines in the newborn neurons. Intriguingly, animals with a double-hit, i.e. injury and ApoE conditionally inactivated in astrocytes, demonstrated the most pronounced impairments in the hippocampal-dependent Morris water maze test, failing to exhibit spatial memory after both acquisition and reversal training trials. In comparison, conditional knockout mice without injury displayed impairments but only in the reversal phase of the test, suggesting accumulative effects of astrocytic ApoE deficiency and traumatic brain injury on AD-like phenotypes. Together, these findings demonstrate that astrocytic ApoE is required for functional injury-induced neurogenesis following traumatic brain injury.

[1] Department of Pediatrics, Columbia University College of Physicians and Surgeons, New York, NY 10032, USA. [2] The Mouse Neurobehavior Core. Institute of Genomic Medicine, Columbia University College of Physicians and Surgeons, New York, NY 10032, USA. [3] Department of Neurology, Columbia University College of Physicians and Surgeons, New York, NY 10032, USA. ✉email: ty2255@cumc.columbia.edu; sk3516@cumc.columbia.edu

The presence of specific human apolipoprotein E (ApoE) isoforms is the best-known risk factor for developing late-onset Alzheimer's disease (LOAD), although the mechanism underlying its role is unknown. In humans, polymorphic ApoE protein is derived from three alleles, E2, E3, and E4. Approximately one-quarter of all individuals are E4 carriers, and 65–80% of LOAD patients have at least one copy of the E4 allele[1]. Despite its clinical relevance, the function of ApoE in the brain remains largely unknown, but is believed to be critical for repairing and remodeling lipid membranes, organelle biogenesis, and neuronal dendritogenesis because of its role in cholesterol and lipid transport and metabolism[1,2].

We have recently demonstrated that systemic and developmental deficiency of ApoE results in a simplification of dendritic tree structure from newborn neurons in the dentate gyrus during both normal development and following controlled cortical impact injury (CCI). In addition, similar phenotypes were observed in humanized ApoE4 replacement mice[3,4]. Moreover, the lack of ApoE, or the presence of human ApoE4, resulted in lower spine density, specifically in newborn neurons in the dentate gyrus. Thus, we demonstrated that ApoE is a prerequisite for both the proper formation of dendritic trees and the spine density of adult hippocampal newborn neurons in naive and injured brains.

In the brain, although ApoE is observed in several types of cells, including astrocytes, microglia, and stressed neurons; the primary source of ApoE is astrocytes[5]. ApoE released from astrocytes forms HDL-like lipoprotein particles with cholesterol and phospholipids[6]. The ApoE-containing lipoprotein particles are then taken up by cells through the LDL receptor family, including LDLR and LDLR-related protein 1 (LRP1) for ApoE-mediated lipid metabolism[6]. In addition, ApoE-containing lipoprotein particles bind with neuronal amyloid β (Aβ), which then undergoes endocytosis by astrocytes facilitating Aβ clearance[7].

To further illustrate the critical roles of astrocytic ApoE in lipid metabolism, neuronal plasticity, and repair, we generated ApoE-conditional mice to spatially and temporally regulate ApoE expression in astrocytes using Aldh1l1-cre/ERT transgenic mice[8]. In this study, we employed our ApoE-conditional mice to specifically explore the role of astrocyte-derived ApoE in maintaining dendritic structure in both pre-existing and newly generated dentate gyrus neurons. In addition, we determined the behavioral outcomes of astrocytic ApoE deficiency and controlled cortical impact (CCI) injury a well-established experimental model of traumatic brain injury (TBI) and a clinically relevant environmental insult commonly associated with increased risk of developing AD[9,10]. Our results reveal that the injury was the major factor causing deficits in dendritic tree complexity, the total length of dendrites, and nodes in pre-existing neurons. However, those impairments in pre-existing neurons were not observed in the newborn neurons. Intriguingly, the reduction of astrocytic ApoE resulted in a significant reduction in spine density in newborn neurons in the dentate gyrus. Although it is not clear whether these morphologic changes cause the observed functional impairments, the spatial memory deficits in ApoE conditionally deficient mice with injury, in both the acquisition and reversal phases of the Morris water maze test, suggest a critical role of astrocytic ApoE. Hence, our findings suggest that astrocytic ApoE is critical for the development of newly born granular neurons in the dentate gyrus, and this effect is particularly evident in the injured brain.

## Results

### Generation of ApoE-conditional knockout animal and experimental design.
In the brain, ApoE is synthesized mainly in astrocytes[2]. In order to investigate its role specifically in astrocytes, we developed a conditional knockout model of ApoE using a Cre-loxp strategy. ApoE$^{f/f}$ mice were derived using a construct in which the first and second exons of the murine ApoE gene were flanked with loxP sites ("Methods", Fig. 1a). To verify that mice had germline transmission of ApoE$^{f/f}$, primers targeting exon 2 of the ApoE gene and upstream loxP site were used to create DNA fragments by performing traditional PCR. The ~700-bps fragment indicated the ApoE$^{f/f}$ mouse whereas the ~500-bps fragment the ApoE$^{wt/wt}$ mouse, and the ~500/700 fragments were found in heterozygous ApoE$^{f/wt}$ mice (Fig. 1b and Supplementary Fig. 1).

ApoE expression has been observed in astrocytes, microglia, and stressed neurons in the brain, though astrocytes are the primary source[5,11,12]. To investigate the specific role of astrocytic ApoE in the brain, we used Aldh1l1-cre/ERT transgenic mice with validated specificity for cre/ERT-dependent recombination in astrocytes[8]. Aldh1l1-cre/ERT transgenic mice were mated with ApoE$^{f/f}$ mice to allow temporally specified deletion of the ApoE gene in astrocytes, and tdTomato-expressing reporter mice Ai14 were used for visualization. At three weeks of age, ApoE$^{f/f}$ and littermate control mice received one injection of tamoxifen each day for 3–5 days, to trigger deletion of ApoE in astrocytes when astrogliogenesis completed and to maximize the possible observable phenotypes (Fig. 1c). When the treated mice were 6-weeks old, they received either a sham operation or CCI; and a subset of them received an intracranial injection of eGFP-expressing retrovirus to label dividing newborn neurons in the dentate gyrus to test if the previously observed phenotypes were recapitulated[3,4]. They were perfused 4 weeks after surgery to analyze the morphology of eGFP-expressing mature newborn neurons (Fig. 1c). For mice not infected with the retrovirus, brains were harvested four weeks after surgery for Golgi–Cox staining to determine the dendritic complexity of neurons in the dentate gyrus (Fig. 1c). Finally, for behavioral testing, habituation started 4 weeks after surgery and was performed once a day for 5 days. Following habituation, elevated plus-maze and open-field tasks were performed to determine whether the lack of astrocytic ApoE affected stress and anxiety. To investigate if the reduction of astrocytic ApoE impaired learning and memory, Morris water maze, and contextual fear-conditioning tasks were performed (Fig. 1c).

### Aldh1l1-cre/ERT induced knockout of ApoE in astrocytes.
To investigate the role of astrocytic ApoE in neuronal plasticity and maturation of newborn neurons in the dentate gyrus in both naive and injured brains, ApoE-conditional knockout mice were crossed with Aldh1l1-cre/ERT BAC transgenic mice to induce astrocyte-specific cre-dependent gene recombination to allow ablation of ApoE expression in astrocytes in a tamoxifen-inducible manner[8]. The mice were also crossed with Ai14, to verify the efficiency of cre-mediated recombination by allowing tdTomato expression where this occurs. Mice received one injection of tamoxifen daily for 3–5 days. Samples of the hippocampus were freshly harvested (Ctr = 4, cKO = 4) 7 weeks after tamoxifen treatments for quantitative RT-PCR to measure the relative amounts of ApoE mRNA in the hippocampus before the performance of behavioral tests. Using GAPDH to normalize the amount of expressing ApoE mRNA, a significant reduction was observed in the hippocampus in conditional knockout mice compared with controls (Fig. 2a, unpaired $t$ test, $t = 6.181$, $P = 0.0008$). To determine the efficiency of ApoE deletion in astrocytes in both naive and injured mice, tamoxifen-treated mice received a sham operation or CCI at 6 weeks of age (Ctr-Sham=3; Ctr–CCI = 3; cKO-Sham=3; cKO–CCI = 3; Fig. 1a). Four weeks after the surgery, brain sections were collected for ApoE staining. Most of the ApoE-expressing cells in the hippocampus expressed tdTomato- from sham-operated and injured non-ApoE$^{f/f}$ siblings, and the morphology of these cells confirmed that they were astrocytes (Fig. 2c–e, arrows). As compared to controls, the number of

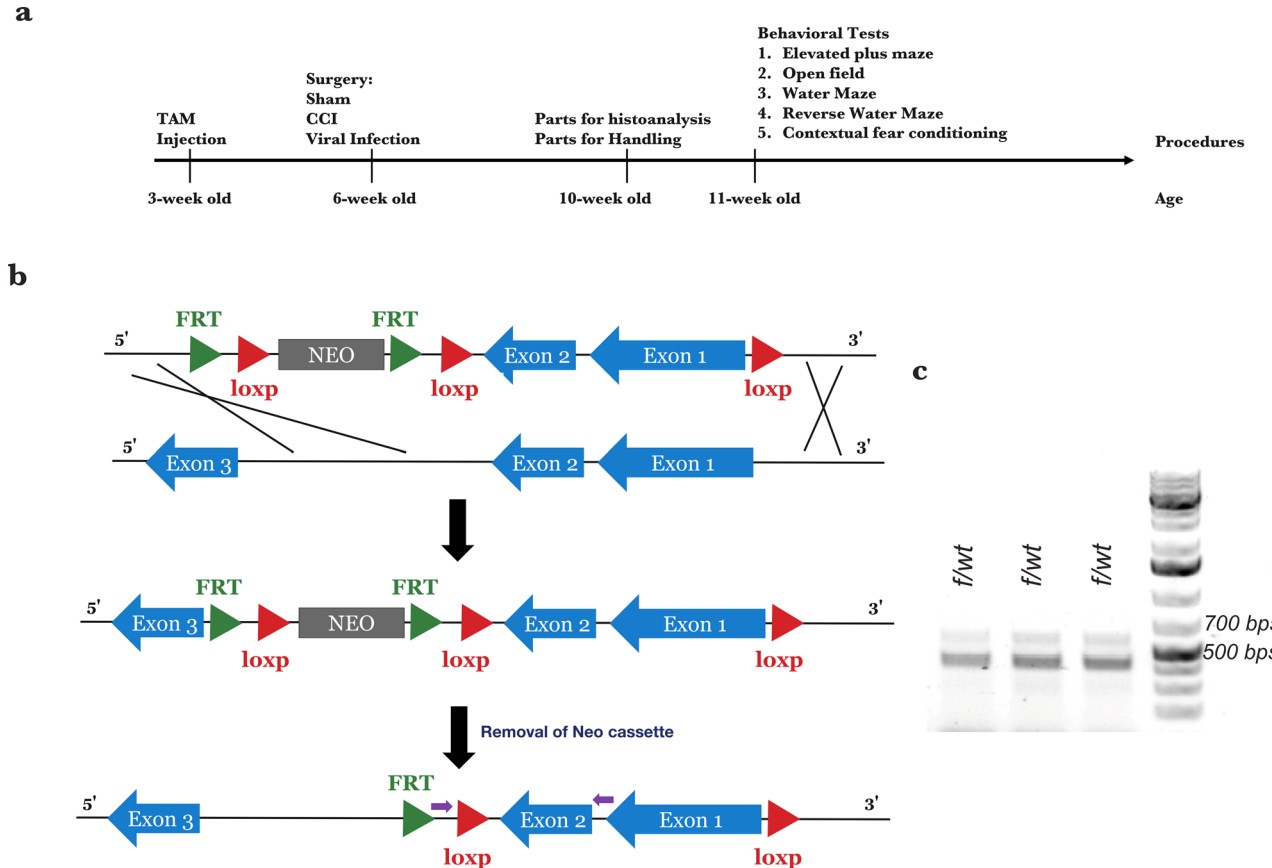

**Fig. 1 Overview of the generation of ApoE-conditional knockout animals and experimental outline. a** To generate ApoE-conditional knockout mice, exon 1 and exon 2 of the ApoE gene were flanked with loxp sites to allow for cre-dependent recombination. The primers for genotyping were designed to flank exon 2 of the ApoE gene with or without one of the loxp sites. **b** Timeline of experiments. Mice received tamoxifen at three weeks of age. At six weeks of age, sham or CCI surgery was performed. A subset of mice received eGFP-expressing retrovirus targeted to the dentate gyrus at the conclusion of surgery. Four weeks after viral infection, mice were perfused to analyze the morphology of newborn neurons or stained using Golgi–Cox method. Habituation was performed on the remaining mice. Following habituation, the elevated plus maze, open field, Morris water maze, and contextual fear-conditioning tasks were performed. **c** The 500 bp DNA fragment generated by PCR indicates mice without the loxp-flanked ApoE, the 750 bp result indicates mice with two copies of loxp-flanked ApoE, and the hemizygous mice generate one 500-bp and one 750-bp fragment.

ApoE-expressing cells in the hippocampus in the ApoE$^{f/f}$ conditional knockout mice was dramatically reduced (Fig. 2f–h, arrows). Using unbiased stereology, $1–1.5 \times 10^6$ ApoE-expressing cells in the hippocampus were counted in both naive and injured controls (Fig. 2b). In the Aldh1l1-cre/ERT; ApoEf/f mice, the number of ApoE-expressing cells was significantly reduced to $0.1–0.5 \times 10^6$ cells in both sham-operated and injured mice without astrocytic ApoE (Fig. 2b, two-way ANOVA, interaction: $F = 0.1232$, $P = 0.7346$; surgery: $F = 1.322$, $P = 0.2835$; genotype: $F = 37.96$, $P = 0.0003$. Sham-Ctr vs Sham-cKO: $P = 0.0144$; Sham-Ctr vs CCI-cKO: $P = 0.0310$; CCI-Ctr vs Sham-cKO: $P = 0.0038$; CCI-Ctr vs CCI-cKO: $P = 0.0076$ in Tukey's post hoc analysis). Therefore, Aldh1l1-cre/ERT induced knockout of ApoE in ~90% of astrocytes in both sham-operated and injured mice. In the absence of the Aldh1l1-cre/ERT transgene, the injection of tamoxifen did not result in the recombination of loxP-flanked genes as no tdTomato-expressing cells were observed and ApoE expression was preserved (Fig. 2i–k, arrows). Hence, Aldh1l1-cre/ERT triggers efficient knockout of the ApoE gene in astrocytes in a tamoxifen-dependent manner.

**Reduction of astrocytic ApoE mildly impairs the dendritic complexity of pre-existing neurons in the dentate gyrus**. As a relay hub, the dentate gyrus receives information from the entorhinal cortex and transmits it back via CA3 and CA1. Adult neurogenesis in the dentate gyrus and the connections between CA3 and the dentate gyrus are known to play a critical role in encoding new memories and distinguishing them from existing ones to prevent memory overlap[13]. The abundant expression of ApoE in the molecular layer of the dentate gyrus where the dendrites of granular neurons reside suggest that ApoE is critical in dendritic formation and maintenance. To elucidate the potential role of astrocytic ApoE in maintaining the dendritic trees of existing neurons in the dentate gyrus in sham-operated and injured mice, Golgi–Cox staining method was deployed to randomly select neurons for analysis. Because it has been estimated that there are ~500,000 mature granular cells in the dentate gyrus and ~2000 newborn neurons and ~7000 in injured brains[14,15], therefore, the Golgi–Cox-stained neurons here were treated as pre-existing neurons. Four weeks after surgery, Golgi–Cox stained neurons in the granular layer of the dentate gyrus were randomly selected for analysis (Fig. 3a, c, e, g. Ctr-Sham = 5 mice, 137 neurons; Ctr–CCI = 4 mice, 372 neurons; cKO-Sham = 4 mice, 100 neurons; cKO–CCI = 4 mice, 260 neurons. Scale bar = 100 μm). The dendritic trees of stained neurons were then restructured for morphological analysis using Neurolucida 360 (Fig. 3b, d, f, h). Using Sholl analysis, two-way ANOVA revealed that injury was the major effect causing impaired distribution of dendrites in the dentate gyrus despise the presence of astrocytic ApoE (Fig. 3i and Supplementary Data 2,

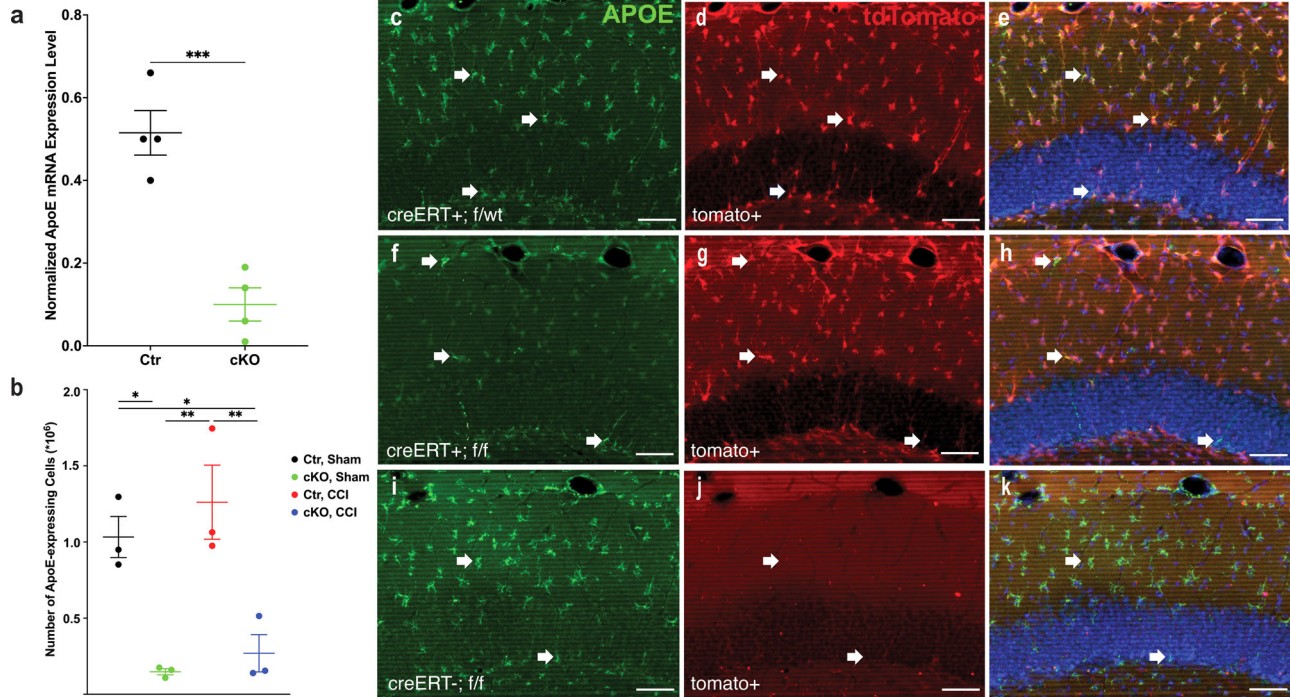

**Fig. 2 Tamoxifen and aldh1l1-creERT-dependent knockdown of ApoE expression in astrocytes. a** Quantitative RT-PCR demonstrates a significant reduction in ApoE mRNA in the hippocampus in conditional knockout mice after treatment with tamoxifen. **b** Significant reduction in the number of ApoE-expressing cells in the hippocampus in aldh1l1-creERT; apoe$^{f/f}$ mice in both sham-operated and CCI mice. **c–e** Seven weeks after tamoxifen in control littermates, ApoE expression was observed in most aldh1l1-dependent tdTomato-expressing cells in the hippocampus, as shown by arrows. **f–h** In apoe$^{f/f}$ mice, ApoE expression in tdTomato-expressing cells is dramatically reduced. **i–k** In the absence of aldh1l1-creERT, the expression of ApoE in the hippocampus remained unchanged in mice receiving tamoxifen and the expression of cre-dependent tdTomato was not observed. This indicates that no spontaneous recombination of loxp-flanked DNA fragments was observed. Data are presented as means ± SEM, *$P < 0.05$; **$P < 0.01$; ***$P < 0.001$. Scale bar = 50 µm.

two-way ANOVA, genotype: $P = 0.345$, $F = 1.086$; surgery: $P < 0.0001$, $F = 3.179$; interaction: $P = 0.045$, $F = 1.489$). In addition, the total lengths and nodes of the dendrites in the existing neurons were impacted by CCI when compared with Ctr-Sham despise the presence of astrocytic ApoE (Fig. 3j, k. For total length: two-way ANOVA, interaction: $F = 0.5788$, $P = 0.4470$; surgery: $F = 26.83$, $P < 0.0001$; genotype: $F = 1.490$, $P = 0.2226$. For node: two-way ANOVA, interaction: $F = 0.9667$, $P = 0.3258$; surgery: $F = 18.51$, $P < 0.0001$; genotype: $F = 0.1408$, $P = 0.7076$).

**Depletion of astrocytic ApoE markedly impairs dendritic complexity of newly born neurons in injured mice.** In order to determine whether ablation of ApoE impairs the development of newly born dentate gyrus neurons, both controls and conditional knockout mice received an eGFP-expressing retrovirus to label dividing newborn neurons at the time of surgery (Fig. 4a. Ctr-Sham = 4 mice, 212 neurons; Ctr–CCI = 5 mice, 133 neurons; cKO-Sham = 5 mice, 197 neurons; cKO–CCI = 4 mice, 214 neurons. Scale bar = 100 µm.). Analysis took place four weeks later to allow for infected newborn neurons to mature (Fig. 4a–d). Unlike what was observed in the existing neurons, the structure of dendrites in newborn neurons was not affected by either injury or the reduction of astrocytic ApoE. However, the combination of the two factors did result in a significant impairment in the dendritic trees of the newborn neurons (Fig. 4e and Supplementary Data 3. Two-way ANOVA, genotype: $P = 0.095$, $F = 1.375$; surgery: $P = 0.050$, $F = 1.491$; interaction: $P = 0.006$, $F = 1.828$). Interestingly, both CCI and astrocytic ApoE did not have effects on the total length and node (Fig. 4f, g. For total length: two-way ANOVA, interaction: $F = 4.434$, $P = 0.0356$;

surgery: $F = 0.6984$, $P = 0.4036$; genotype: $F = 0.4995$, $P = 0.4800$. For node: two-way ANOVA, interaction: $F = 1.877$, $P = 0.1771$; surgery: $F = 0.8648$, $P = 0.7688$; genotype: $F = 0.7513$, $P = 0.7513$). Our previous studies demonstrated that ApoE also played a critical role in spine density[3,4]. Here, we tested whether the observed deficits in spine density is caused by astrocytic ApoE. As expected, the knockdown of ApoE expression in the astrocytes resulted in downregulation of spine density in newborn neurons in both sham-operated (Fig. 5a, b, e) and injured mice (Fig. 5a–e. Ctr-Sham: 88 dendrites in three mice. cKO-Sham: 91 dendrites in four mice. Ctr–CCI: 166 dendrites in three mice. cKO–CCI: 200 dendrites in four mice. Two-way ANOVA, interaction: $F = 3.221$, $P = 0.0733$; surgery: $F = 0.4179$, $P = 0.5183$; genotype: $F = 259.3$, $P < 0.0001$. ****$P < 0.0001$ in Tukey's post hoc analysis). Thus, we conclude that astrocytic ApoE is critical for the development of the spine density of newborn neurons in both sham and injured mice.

**Spatial memory was impaired in ApoE cKO mice and worsened by CCI.** In the Morris water maze test, CCI did not impair spatial memory in control (Ctr) mice, either in acquisition or reversal (Fig. 6a, b). Using One-Way RM ANOVA followed by Dunnett's test for post hoc comparisons, we found that both Ctr-Sham and Ctr–CCI mice spent significantly more time in the trained quadrant than in the other three quadrants in the probe trial following acquisition training (the main effect of "quadrant": Ctr-Sham, $F_{3,13} = 31.55$, $P < 0.0001$; Ctr–CCI, $F_{3,14} = 8.22$, $P = 0.0019$). Post hoc comparisons indicate that both Ctr-Sham and Ctr–CCI spent significantly more time in the trained quadrant than in the other three quadrants (in Ctr-Sham: $P < 0.0001$ vs

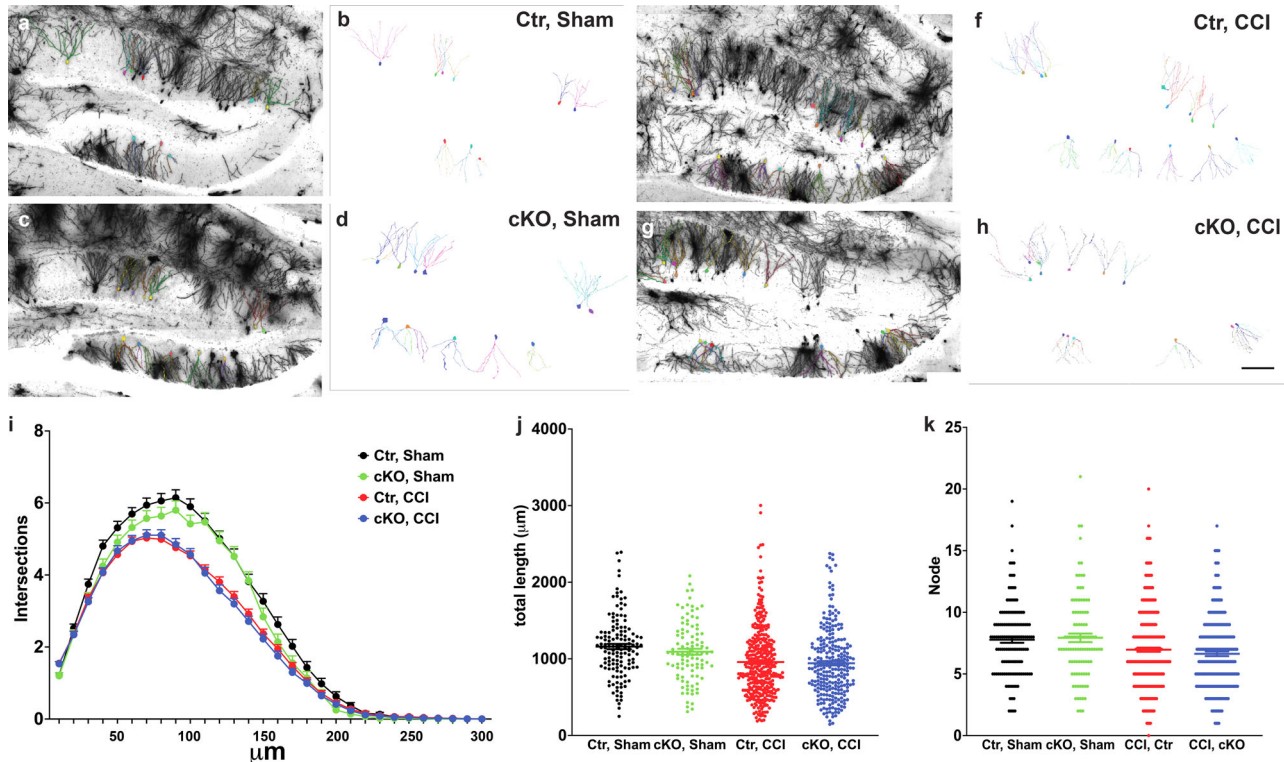

**Fig. 3 The morphology of Golgi–Cox-stained neurons in the hippocampus mice without astrocytic ApoE. a–h** The left panels demonstrate Golgi–Cox-stained neurons in the dentate gyrus and the right panel depicts individual analyzed neurons. **i** The dendritic trees of the existing neurons in the dentate gyrus were impaired significantly in CCI groups but were not affected by the reduction of astrocytic ApoE. **j, k** CCI resulted in shorter total length and fewer nodes in the existing neurons in the dentate gyrus despite of the genotype. Data are presented as means ± SEM. ****$P < 0.0001$. Scale bar in (**h**) = 100 μm.

left, $P = 0.0001$ vs right, and $P < 0.0001$ vs opposition; in Ctr–CCI: $P = 0.0020$ vs left, $P = 0.0215$ vs right, and $P = 0.0296$ vs opposition). Similarly, in the probe trial following reversal training, both Ctr-Sham and Ctr–CCI mice spent significantly more time in the trained quadrant (the main effect of quadrant: Ctr-Sham, $F_{3,13} = 7.06$, $P = 0.0021$; Ctr–CCI, $F_{3,14} = 4.69$, $P = 0.0137$). Post hoc comparisons indicated that the differences between the trained quadrant and all other three quadrants were significant for Ctr–CCI ($P = 0.0402$ vs left, $P = 0.0243$ vs right, and $P = 0.0402$ vs opposition). For Ctr-Sham, the differences between the trained quadrant and the quadrants opposite of, and right to, the trained quadrants were significant ($P = 0.0971$ vs left, $P = 0.0060$ vs right, and $P = 0.0211$ vs opposition).

In ApoE-conditional knockout mice (cKO), an additive effect of ApoE deficiency and CCI was seen, with uninjured cKO mice only exhibiting impaired memory following the acquisition, and injured cKO mice exhibiting impaired memory following both acquisition and reversal. While cKO-Sham mice exhibited a clear preference for the trained quadrant in the probe trial following acquisition ($F_{3,13} = 7.53$, $P = 0.0040$), spending significantly more time in the trained quadrant than in the other three quadrants ($P = 0.0270$ vs left, $P = 0.0255$ vs right, and $P = 0.0207$ vs opposition), this group failed to exhibit quadrant preference in the probe trial following reversal training ($F_{3,13} = 2.32$, NS), indicating modest impairment. A more pronounced impairment was seen in cKO mice with CCI, which did not exhibit a preference for the trained quadrant in the probe trials after acquisition training ($F_{3,14} = 0.85$, NS) or the reversal training ($F_{3,14} = 3.09$, NS).

Latency to the platform, swim distance and swim speed were analyzed by genotype (Ctr and cKO) (Figs. 6c–f and 7a–d). Two-way RM ANOVA was used, with "Day" as the within-subject factor, and operation (Sham vs. CCI) as the between-subject factors. During acquisition training days, latency to the platform was significantly longer in Ctr–CCI mice than in Ctr-Sham mice ($F_{1,27} = 10.00$, $P = 0.0038$). No significant effect of CCI was found for latency to the platform in cKO mice ($F_{1,25} = 2.53$, NS). During reversal training days, no significant effects of CCI were found on latency to the platform in Ctr mice ($F_{1,27} = 1.22$, NS) or cKO groups ($F_{1,25} = 0.23$, NS).

The increase in latency in the Ctr–CCI group is probably related to a decrease in swim speed in this group (Fig. 7c, d). Compared to Ctr-Sham mice, swim speed was significantly slower in Ctr–CCI mice during acquisition ($F_{1,27} = 4.47$, $P = 0.0439$) and reversal ($F_{1,27} = 9.75$, $P = 0.0042$). In cKO mice, swim speed was slower in cKO–CCI mice during acquisition ($F_{1,25} = 5.59$, $P = 0.0261$). No speed difference was found in reversal ($F_{1,25} = 1.17$, NS). In Ctr mice, swim distance was shorter in Ctr–CCI mice than in Ctr-Sham mice during acquisition ($F_{1,27} = 9.05$, $P = 0.0056$) but not during reversal ($F_{1,27} = 0.05$, NS). In cKO mice, swim distance was not different between the Sham and CCI group during acquisition ($F_{1,25} = 0.39$, NS) or reversal ($F_{1,25} = 0.23$, NS). In sum, Ctr mice exhibited normal learning and memory in the water maze acquisition and reversal, despite CCI's effect in reducing swim speed. In contrast, a partial deficit was found in cKO-Sham mice, which exhibited normal acquisition memory but impaired reversal memory. This deficit was exacerbated by CCI, as evidenced by cKO–CCI mice showing impaired memory after both acquisition and reversal learning trials.

In the fear-conditioning test (Fig. 8d–g), no significant differences were found in post-training freezing. In the context conditioning test on Day 2, a significant operation effect was found ($F_{1,52} = 5.14$, $P = 0.0275$) with CCI mice showing reduced freezing than animals with the sham operation. Genotype ($F_{1,52} = 0.68$, NS) and genotype x operation interaction

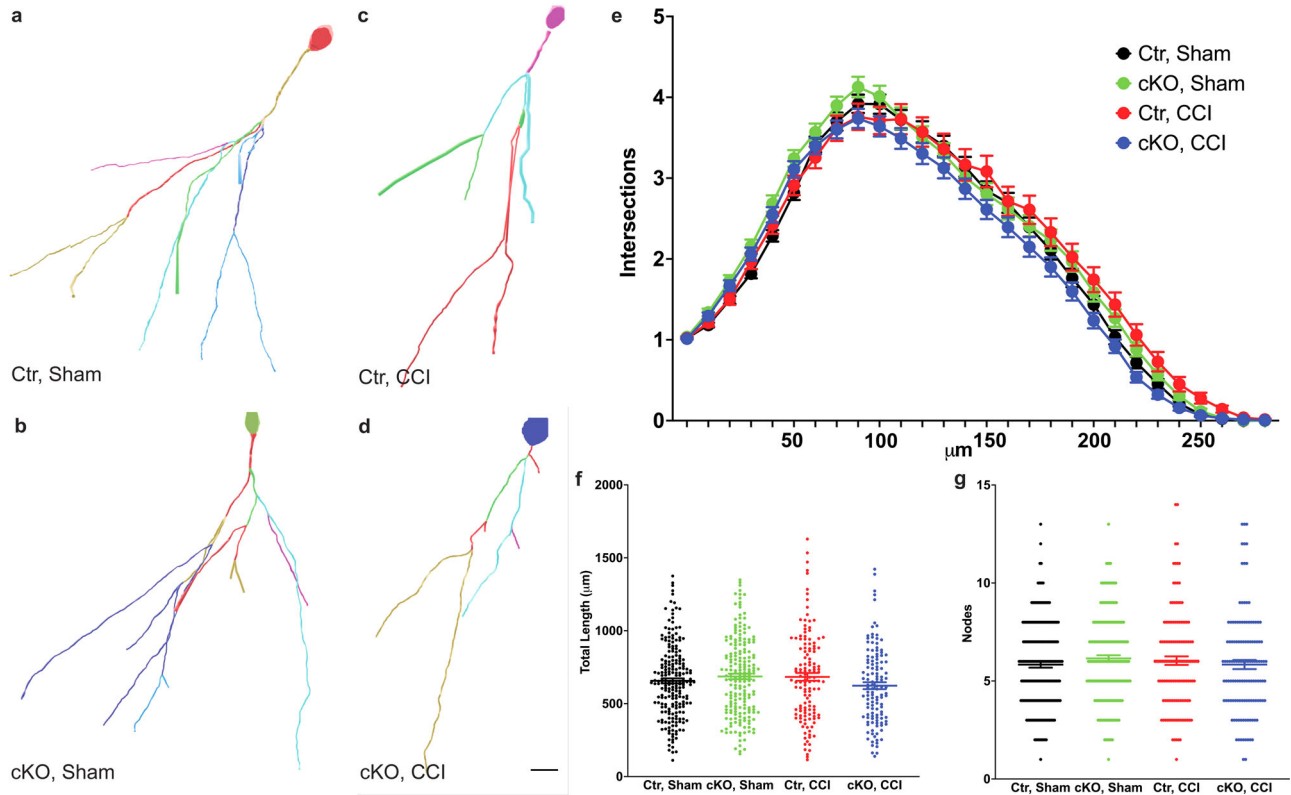

**Fig. 4 The reduction in astrocytic ApoE impairs dendritic complexity in newborn neurons in the injured dentate gyrus. a–d** Reconstructed dendritic trees of newborn neurons 4 weeks after being injected with a GFP-expressing retrovirus in the dentate gyrus. **e** Neither the reduction of astrocytic ApoE nor CCI results in deficits in dendrites of newborn neurons in the dentate gyrus. **f, g** Similar observations were seen in the total length and nodes in the newborn neurons. Scale bar in (**d**) = 10 μm.

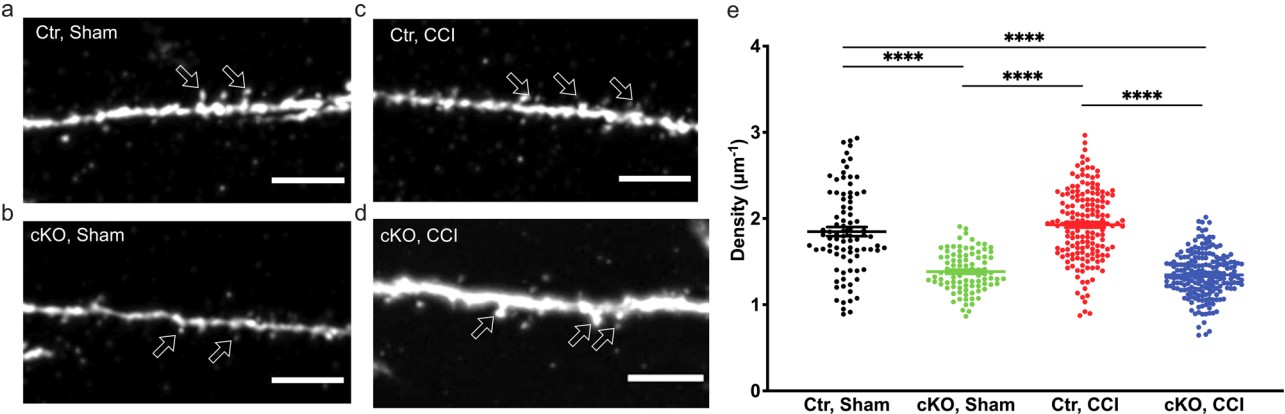

**Fig. 5 Lack of astrocytic ApoE results in the reduction of spine density in newborn neurons. a–d** Representative images illustrate the distribution of the spines in the dendrites in newborn neurons. **e** The reduction of astrocytic ApoE was the main factor, not CCI, that resulted in reduced spine density in newborn neurons. Arrows indicate the spines. Data are presented as means ± SEM. ****$P < 0.0001$. Scale bar = 5 μm.

($F_{1,52} = 0.28$, NS) were not significant. No significant differences were found in pre-cue freezing or cued freezing on day 3.

**Anxiety-like behavior and exploratory activity.** As illustrated in Fig. 8a, b, in the elevated plus-maze test, two-way ANOVA revealed a significant effect of genotype on % time in the open arms, which is reduced in cKO mice ($F_{1,48} = 7.88$, $P = 0.0072$). The main effect of operation ($F_{1,48} = 1.48$, NS) and the interaction between genotype and operation ($F_{1,48} = 2.59$, NS) were not significant. For total arm entries, no significant effects were found for genotype ($F_{1,48} = 0.22$, NS), operation ($F_{1,48} = 0.00$, NS), or

the genotype x operation interaction ($F_{1,48} = 2.56$, NS). In the open-field test (Fig. 8c), two-way ANOVA revealed a significant effect of operation, with CCI mice showing increased distance traveled in the open field ($F_{1,52} = 4.11$, $P = 0.0477$). Effects of genotype ($F_{1,52} = 0.59$, NS) and genotype x operation interaction ($F_{1,52} = 0.25$, NS) were not significant. Results of the open-field test ruled out gross motor impairments in cKO mice.

## Discussion
ApoE is abundant in the brain and is primarily synthesized by astrocytes, yet its role in neuronal development and maintenance

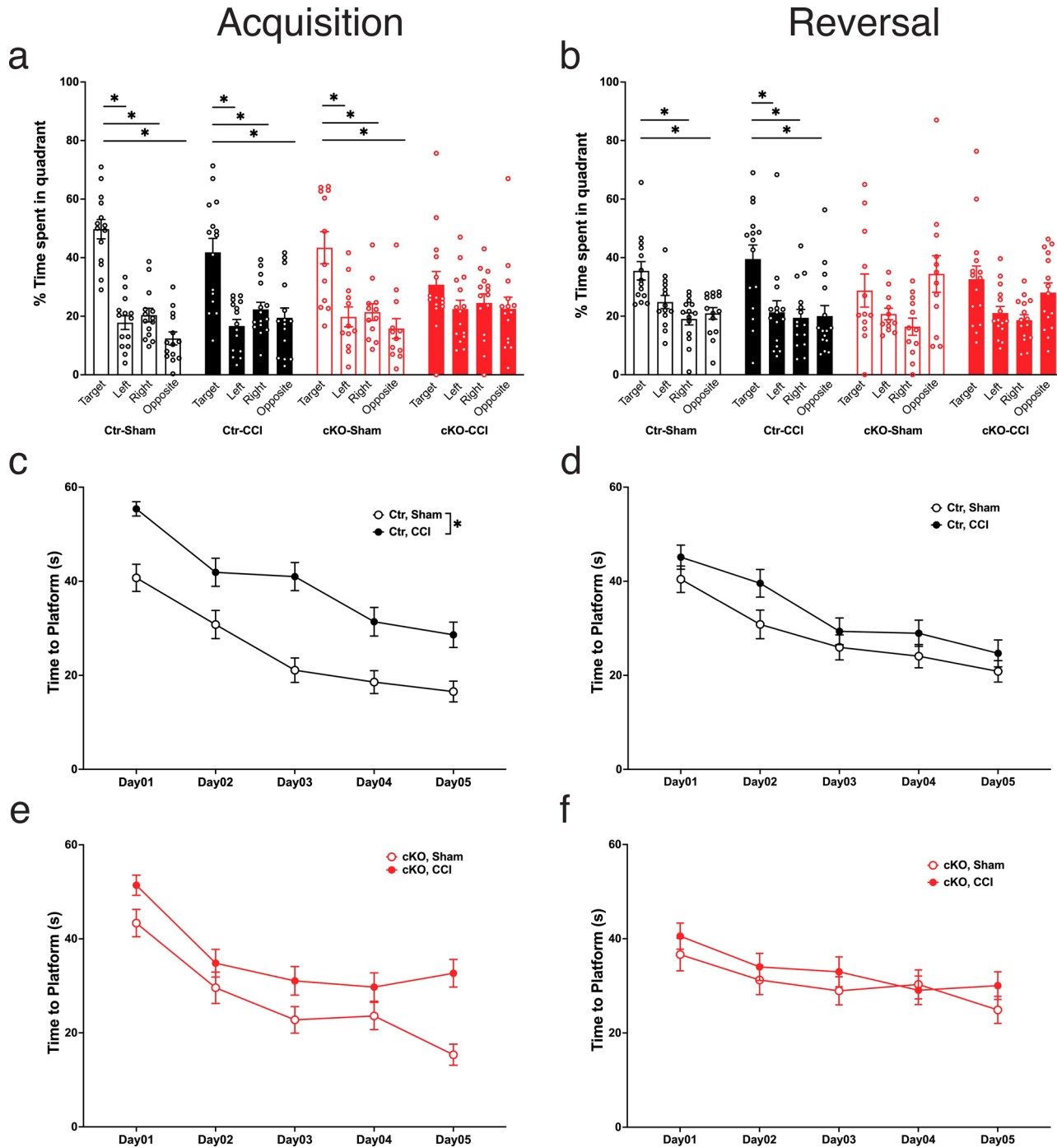

**Fig. 6 Impaired spatial learning in astrocytic ApoE-deficient mice was worsened by CCI in the Morris water maze test. a** Astrocytic ApoE deficiency did not affect spatial memory in uninjured mice after acquisition training but did impair memory in cKO mice with CCI. **b** Astrocytic ApoE deficiency alone was sufficient to impair reversal memory. The same impairment was seen in cKO mice with or without CCI. **c, e** During the acquisition training session, injured mice exhibited longer latency to the platform than sham-operated mice. This was significant in controls and approaching significance in cKO mice. **d, f** During the reversal training session, no difference was observed. Data are presented as means ± SEM. *$P < 0.05$.

remains surprisingly unclear. To determine how astrocytic ApoE might affect adult neurogenesis, we generated conditional ApoE knockout mice and crossed them with well-established astrocyte-specific Aldh1l1-creERT mice to ablate the expression of ApoE in astrocytes[8]. After tamoxifen treatment, we observed an ~90% reduction of ApoE in the hippocampus. This decrease in astrocytic ApoE resulted in mild impairments in the complexity of the existing dendritic tree and more pronounced ones in newborn granular cells. This effect was most evident in injury-induced

neurons in the dentate gyrus. This attenuation in granular cell complexity may underlie impairments in reversal learning observed in the Morris water maze. Though mice learned to find the hidden platform during the training session, ApoE-deficient mice failed to retrieve the platform's location in the reversal learning task. A controlled cortical impact model of traumatic brain injury augmented the deficits observed in the sham groups. Thus, the lack of astrocytic ApoE simplified the structure and shortened the total length of the dendritic trees of newborn

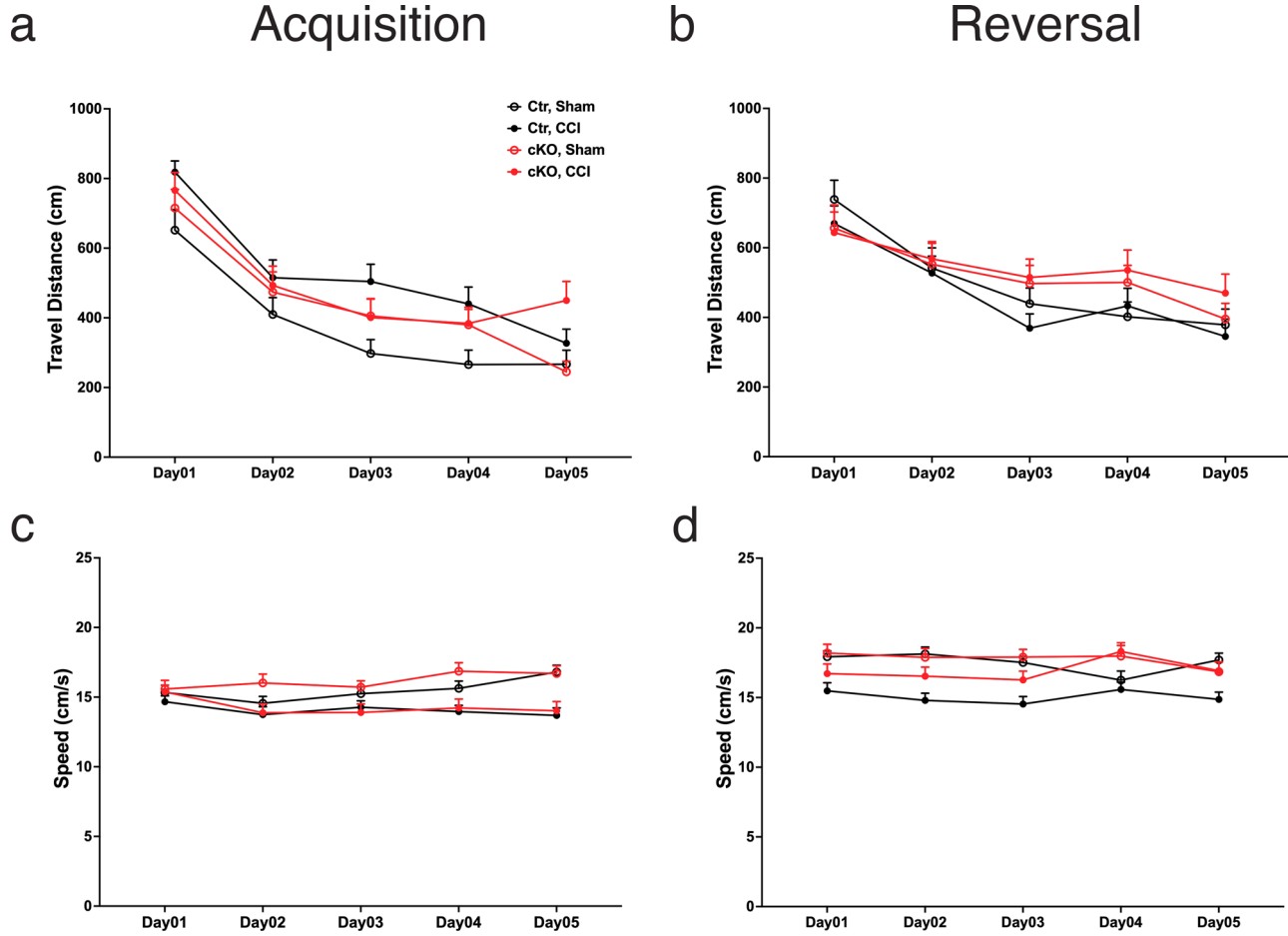

**Fig. 7 Water maze speed and distance were modestly affected by CCI but not by ApoE deficiency. a**, **c** During acquisition, injured mice swam longer distances to reach the hidden platform as compared to the control groups, however, there was no significant CCI effect on swim distance in mice lacking astrocytic ApoE. The injury also resulted in slower swimming speed despite the presence of astrocytic ApoE. **b**, **d** In reversal learning trials, all mice swam similar distance to reach the new position of the platform. Interestingly, the swimming speed was slower in Ctr–CCI group. However, there was no difference observed between cKO-Sham and cKO-CCI. Data are presented as means ± SEM.

neurons in the dentate gyrus and impaired acquisition and reversal memories.

ApoE is ubiquitously expressed in the brain. Using mice with the ApoE locus replaced by eGFP, ApoE expression was seen in 75% of astrocytes, 10% of microglia, and essentially no neurons[5]. However, additional studies have demonstrated ApoE in pericytes, oligodendrocytes, choroid plexus, and adult neural progenitors[2,16–20]. In addition to astrocytes, the adult neural progenitor population has also been shown to express Aldh1l1[21]. To distinguish its function in the present study, tdTomato-expressing granular cells in the dentate gyrus were only occasionally observed using Ai14 mice to trace the Aldh1l1 active cells via expression of tdTomato. Thus, we achieved efficient and specific ablation of ApoE expression in astrocytes with a minimal deletion in the stem cells themselves, where ApoE has been shown to be expressed and required for progenitor proliferation[16]. A recent study also demonstrated that Aldh1l1-creERT mice can be used to study the functions of non-neural progenitors astrocytes[22]. However, the activity of Aldh1l1 promotor in the neural progenitors in young adults needs to be studied in a more comprehensive manner.

We utilized the Golgi–Cox staining method to investigate whether astrocytic ApoE affected the existing dendritic structure in the dentate gyrus. In the sham-injured groups, a small but statistically significant difference in the dendritic structure was observed using Sholl analysis between controls and astrocyte-

specific ApoE knockout mice. In previous studies using constitutive ApoE-deficient mice and MAP2 as a marker of hippo-campal dendritic trees, lack of ApoE resulted in a decrease of neuropil in the inner molecular layer of the dentate gyrus starting from 4-months of age[23]. However, using the same approach, others reported that no significant difference was observed in the dentate gyrus[24]. Furthermore, conflicting results were reported using a transgenic approach to overexpress human ApoE driven by cell-specific promoters or using humanized ApoE mice[25–27]. Using human ApoE4-targeted replacement mice, dendritic trees were less complex compared with wild-type mice, although the structure in the dentate gyrus was not studied[26]. When using a cell-specific promoter to drive the expression of ApoE, over-expression of human ApoE4 in neurons resulted in few dendrites in CA1 neurons, but no deficit was observed when overexpressed in astrocytes[27]. The caveats in these published reports include poor control of the expression level of ApoE when using a transgenic approach and the developmental effects when using constitutively ApoE-deficient mice. The conditional knockout approach provided in this study provides a more specific and precise manner to study the functional role of ApoE in the brain.

Using the targeted replacement of human ApoE with mouse ApoE, and ApoE-deficient mice, we previously demonstrated its requirement for neuronal maturation in both sham-operated and injured brains[3,4]. In ApoE-deficient mice and humanized ApoE4 mice, newborn neurons showed less complex dendritic trees and

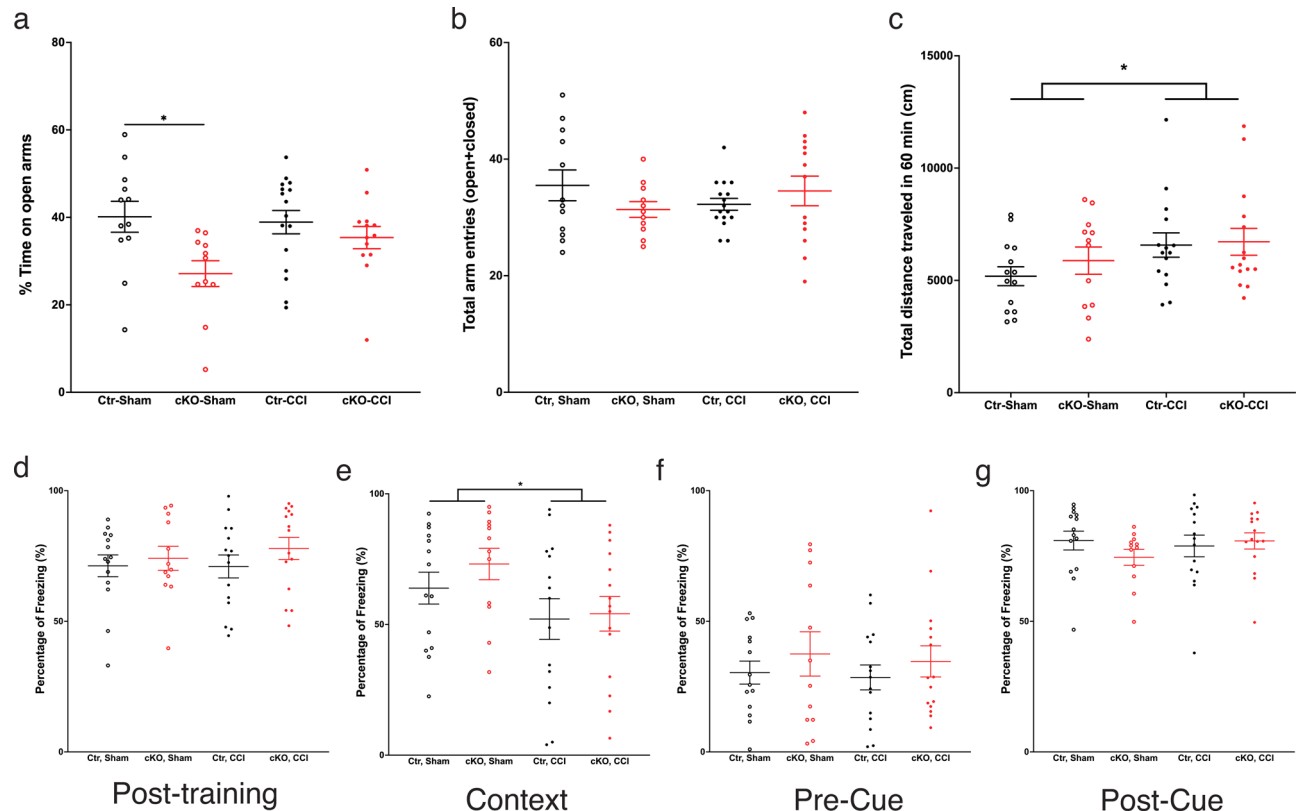

**Fig. 8 Elevated plus maze, open field, and fear conditioning. a, b** In the elevated plus-maze task, the cKO-Sham mice spent significantly less time in the open arm compared with Ctr-Sham. No difference was observed between Ctr–CCI and cKO–CCI. For total entries of arms, no difference was observed among the four groups. **c** In the open-field test, CCI mice traveled significantly longer distance compared with sham-operated mice, indicating CCI does not impair locomotor activity. **d–g** In the fear-conditioning test, no differences were observed during post-training, indicating similar responses to pain (**d**). In day 2 contextual memory test (**e**), CCI mice exhibited reduced % time freezing as compared with Sham groups. In the day-3 cued-memory test (**f**, **g**), no differences were observed among the four groups. Data are presented as means ± SEM. *$P < 0.05$.

less spine density when compared with wild-type mice under normal as well as a pathological condition. Adult neurogenesis in the dentate gyrus is facilitated by astrocytes, microglia, and neurons, and ApoE is synthesized in astrocytes and microglia. By specifically deleting ApoE in astrocytes after the majority of brain development has occurred, we observed that the development of dendrites of the newborn neurons were not affected by either the reduction of astrocytic ApoE or CCI. However, the combination of two factors did affect the development. Consistent with the previous study, newborn neurons in injured mice grew less spine density when lacking ApoE[4]. Thus, astrocytic ApoE is critical in spine formation in newborn neurons in the dentate gyrus. It is also known that traumatic brain injury predisposes individuals towards Alzheimer's disease and other neurodegenerative diseases later in life[1,28–30]. Interestingly, impaired neurogenesis is known to accelerate the progression of neurodegenerative diseases[31,32]. The present study suggests that astrocytic ApoE may play a role in the progression of neurodegenerative disease resulting from impaired hippocampal newborn neurons.

To investigate if the lack of astrocytic ApoE affected learning and memory or anxiety-like behaviors, a robust battery of behavioral tests was performed in this study. Intriguingly, the lack of ApoE impaired reversal learning and memory in the Morris water maze task, indicating compromised cognitive flexibility[33,34]. Despite the lack of astrocytic ApoE, mice exhibited intact spatial memory after acquisition training. In comparison, injured mice learned more slowly compared to sham-operated mice. In the memory test after acquisition training, sham-operated mice spent more time in the target quadrant regardless of whether astrocytic

ApoE was present or not. In the memory test after reversal training, astrocytic ApoE-deficient mice exhibited impaired spatial memory. A more pronounced impairment was seen in mice with double insults: absence of astrocytic ApoE and CCI. This group exhibit impaired spatial memory after both acquisition and reversal. While we did observe mildly reduced swim speed in CCI groups, it is unlikely that this reflects gross motor impairments, which could have confounded the water maze results. In the open-field test, CCI groups actually transversed greater distance than uninjured mice, indicating that CCI did not impair mobility.

The Morris water maze is a well-established behavioral task that critically depends on the hippocampus to succeed in spatial learning and memory in laboratory rodents. For the reference memory version of the Morris water maze task, experimental animals learn to find the hidden platform according to the surrounding environmental cues in the task room for task acquisition. The success of acquisition is later defined by allowing the animal to swim freely in the water tank without the platform and measuring the spent time in the target area. By lesioning specific brain areas, it has been shown that an intact hippocampus is critical for accomplishing this spatial task[33,34]. Moreover, various brain injuries, including TBI, that result in a damaged hippocampus due to secondary injury also cause different levels of impairments in the Morris water maze task in a hippocampal neurogenesis-dependent manner[15]. This study demonstrates a correlation between the astrocytic ApoE and the performance in the Morris water maze. Furthermore, the reversal learning of the Morris water maze revealed that without astrocytic ApoE, both sham-operated and injured mice failed to remember the new

location of the platform and spend significant time in the previous target area. Whether the poor performance in reversal Morris water maze is correlated with impaired newborn neurons remains unclear. Immediate-early genes have been used to identify neurons that associate with a task[35]. Using c-fos as a marker to identify the active neurons and BrdU, to label the newborn neurons, it has been demonstrated that the newborn neurons are preferentially recruited in spatial memory[36]. Furthermore, c-fos has been used to determine the brain-wide memory network[37]. Applying these and other more precise molecular tools, the role of astrocytic ApoE in the functional recruitment of newborn neurons in certain behavioral tasks and brain-wide network can be more clearly elucidated.

Overall, it appears that adult-generated dentate gyrus neurons require ApoE not only for proper dendritic structural maturity but also to regulate certain aspects of critical memory formation. The requirement for ApoE is particularly evident in the setting of traumatic brain injury and suggests the importance of future investigation particularly regarding the cell-specific function of human isoforms of ApoE that have long been correlated with impaired outcomes following TBI through unknown mechanisms. By linking ApoE function with hippocampal-specific recovery following TBI, this present work underlines the importance of additional investigation regarding how ApoE status in humans may influence both recovery and potential treatment pathways for those with both neurodegenerative and trauma-induced brain disease.

## Methods

**Animals**. All procedures in this research were performed following the animal protocol approved by the Institutional Animal Care and Use Committee at Columbia University. Subject mice were housed humanely and cared for inside the barrier facility managed by the Institute of Comparative Medicine at Columbia University. Food and water were provided ad libitum for all mice and were kept on a 12 h light/dark cycle with lights on at 7 AM. B6; FVB-Tg(Aldh1l1-cre/ERT2) 1Khakh/J and Ai14(RCL-tdT)-D lines were ordered from The Jackson Laboratory to allow expression of tdTomato in astrocytes specifically in a tamoxifen-induced manner.

The generation of the ApoE$^{f/f}$ mouse was conducted by the Knockout Mouse Program (KOMP) at the University of California, Davis (project number 456). Briefly, exon 1 and exon 2 of murine ApoE were targeted and flanked with loxp sequences; therefore, the start of transcription is impaired. Upon generation of germline transmitted ApoE$^{f/f}$ mice, they were mated with Tg(Pgk1-flpo)10Sykr mice (Stock No. 011065, The Jackson Laboratory) to remove the neo cassette. These progenies were mated with Aldh1l1-creERT2 and Ai14 to get Aldh1l1-cre/ERT2; Ai14; ApoE$^{f/f}$, littermates with other genotypes were used as controls. To identify the genotype of ApoE$^{f/f}$ mice, primer pairs (forward: 5′-CCGTGCTGTT GGTCACATTGC-3′; reverse:5′-GCATGCACTGTCTTGTATCCTATGTAG-3′) were used to perform PCR. The band with size ~500 bps represents wild-type ApoE gene and ~700 bps PCR fragment indicates the loxp-flanked ApoE gene.

**Controlled cortical impact (CCI) and retroviral injections**. Six-week-old mice were used for the experiments. For controlled cortical impact injury (CCI), we used a standard protocol with a controlled cortical impact device (Leica Impact One, Leica Biosystems) to generate brain injuries[4]. Following anesthesia with isoflurane, mice were placed in a stereotactic frame. A midline incision was made, the soft tissues were reflected, and a $5 \times 5$ mm craniotomy was made between bregma and lambda 1 mm lateral to the midline. The injury was generated with a 3-mm stainless steel tipped impact device with deformation of 0.7 mm, a constant speed of 4.4 m/s, and a duration of 300 ms. Mice operated on but not impacted were used as sham groups. Male and female mice were used for all experiments.

Immediately following the injury (Ctr-Sham=5; cKO-Sham=5; Ctr–CCI = 5; cKO–CCI = 8), 1 µl of a retrovirus with an enhanced green fluorescent protein (eGFP) expressing vector (RV-CAG-eGFP, $1 \times 10^8$ transducing unit/mL, Salk Institute) was infused stereotactically into the dentate gyrus at an injection rate of 0.1 µl per minute by a microinfusion pump (KD Scientific) via a 10 µl microsyringe (Model 801, Hamilton). The following coordinates were used: anterior/posterior: −2.0 mm, medial/lateral: +/−1.55 mm, and dorsal/ventral: −2.0 mm in reference to bregma. After surgery, the scalp was closed with sutures, and mice were placed in their cages and allowed to recover from anesthesia.

**Tissue processing and immunohistochemistry**. Four weeks after surgery, mice were perfused with 4% paraformaldehyde (PFA, 441244, Sigma-Aldrich) and

brains were harvested for post-fixation in 4% PFA overnight. The brains were then sectioned serially using a vibratome (VT1000S, Leica) with a thickness of 50 µm. All sections encompassing the hippocampus were harvested sequentially into six-well plates.

A set of sections was washed with 1×PBS then permeabilized with 0.3% of Triton X-100/1×PBS (PBST) at room temperature. Sections were blocked with normal donkey serum (5% NDS, 017-000-121, Jackson ImmunoResearch). The primary antibodies to label ApoE (1:5000, 7333, ProSci) and eGFP (1:500, A11122, Invitrogen) were added and incubated with sections overnight at room temperature with 0.02% of sodium azide. The following day, sections were washed with PBST, then biotinylated donkey anti-goat IgG was used to label ApoE primary antibody and Alexa 488-conjugated donkey anti-rabbit IgG eGFP antibody (1:200, both from Jackson ImmunoResearch) at 4 °C overnight. On the third day, sections were washed again with PBST, then Alexa 647-conjugated streptavidin was incubated with sections for visualization of ApoE at room temperature for three hours. Then sections were washed with 1×PBS before being mounted on the slides with Vectashield mounting medium (H-1500, Vector Laboratories).

**Quantification of ApoE-expressing cells**. To verify if ApoE expression was suppressed in the brains because of tamoxifen treatment in a cre-dependent manner, ApoE-expressing cells were counted using an unbiased stereological method. Samples (Ctr-Sham=3; cKO-Sham=3; Ctr–CCI = 3; cKO–CCI = 3) were analyzed under a Zeiss microscope (Axio Imager M2, Zeiss). The whole hippocampus was traced under a ×10 objective lens. The sample grids ($500 \times 500$ µm) were determined randomly and cells within the counting frames ($75 \times 75$ µm) were counted under a ×40 objective. To prevent artifacts that resulted from sectioning, a dissector height of 30 µm was used. The ApoE-expressing cell number was estimated using the built-in weighted section thickness method with a coefficient of error less than 0.1 (Stereology Investigator, MBF).

**RNA isolation, reverse transcription, and real-time PCR**. Total RNA was isolated using TRIzol (15596026, ThermoFisher) from the dissected hippocampus after tamoxifen injection as described above (Ctr-Sham=4, cKO-Sham=4). The first-strand cDNA was synthesized using SuperScript First-Strand Synthesis System for RT-PCR with oligo-dT as the primer (18091150, ThermoFisher). Next, the harvested first-strand cDNA was mixed with suitable primers and iTaq Universal SYBR Green Supermix (1725129, Bio-Rad) to perform qPCR using CFX96 Real-Time PCR System (Bio-Rad). The relative amount of ApoE mRNA expression was normalized to the internal control gene, GAPDH. The sequences of used primers for qPCR were: ApoE forward, 5′-TCCTGTCCTGCAACAACATCC-3′; ApoE reverse, 5′-AGGTGCTTGAGACAGGGGC-3′; GAPDH forward, 5′-CCATTCTCGGCCTT GACTGT-3′; GAPDH reverse, 5′-CTCAACTACATGGTCTACATGTTCCA-3′. All primers were synthesized by Integrated DNA Technologies[16].

**Golgi–Cox staining**. Standard Golgi–Cox staining method was followed[38]. Stock solutions of 5% (w/v) of potassium dichromate (P5271, Sigma), mercuric chloride (AC201430250, Fisher Scientific), and potassium chromate (216615, Sigma) were prepared and stored at room temperature in the dark. The working solution was prepared by mixing 50 mL of potassium dichromate with mercuric chloride stock solution, then 40 mL of potassium chromate stock solution was added followed by 100 mL of distilled water. The prepared working solution was left at room temperature covered with foil for at least 48 h to allow for precipitate formation.

Four weeks after surgery (Ctr-Sham=4; cKO-Sham=4; Ctr–CCI = 3; cKO–CCI = 4), mice were sacrificed and brains were removed fresh and washed with distilled water. The washed brains were then transferred into a small bottle that contained 10 mL of clear Golgi–Cox working solution and stored at room temperature. One day later, the solution was renewed and left for 7 days for impregnation. After impregnation, brains were transferred to new bottles containing 30% sucrose solution and stored at 4 °C and protected from light. One day later, the brains were transferred into new bottles with sucrose solution for 4–7 days at 4 °C in the dark.

Brains were cut using a vibratome (VT1000S, Leica) and serial 100-µm sections encompassing the entire hippocampus were harvested. Sections were positioned on gelatin-coated slides to allow further development with 3:1 ammonia solution (1054261000, Millipore Sigma) for 8 min. After development, tissue was dehydrated using serial ethanol solutions. Sections were then kept in Xylene (23400, Electron Microscopy Sciences) before mounting using permount (SP15-500, Fisher Scientific).

**Neuronal morphological analysis**. Sections with either eGFP-expressing cells or Golgi–Cox-stained cells were visualized using a Zeiss microscope (Axio Imager M2, Zeiss) with a Hamamatsu camera (Orca-R2, Hamamatsu). Stack images with 1 µm interval on z axis were acquired (Stereology Investigator, MBF) under a ×20 objective lens. Neurolucida 360 (MBF) was used to reconstruct and analyze the dendritic structure. The total length of dendrites, nodes of dendrites, and Sholl analysis were performed. To quantitate spine density[39], the dendrites in the intact eGFP-expressing cells in the molecular layer of the dentate gyrus were selected and imaged. The image stacks were obtained using a Plan-Apochromat ×63 oil-immersion objective lens (NA 1.4, Zeiss) at the Zeiss Axio Imager M2 microscope

with Apotome 2 with 0.25 μm z axis interval. The resulting scaling per voxel was 0.1 × 0.1 × 0.25 μm. The images were then deconvolved using Autoquant X3 (Media Cybernetics). The dendrites in the molecular layer were traced using Neurolucida 360 (MBF), and spines were identified and counted manually by the experimenter in a blinded manner.

**Behavioral tests.** Four weeks after surgery, subject mice were acclimated to a vivarium room dedicated for housing behavioral subjects for a week, and then a carefully selected test battery was conducted in this order: elevated plus maze, open field, Morris water maze acquisition, Morris water maze reversal, and contextual fear conditioning. Inter-test intervals were 3–7 days. All behavioral experiments were done between 10 AM and 4 PM during the light phase. The experimenter was blind to genotype and injury information while conducting the experiments and subsequent data analysis. Ctr-Sham=14, Ctr–CCI = 15, cKO-Sham=12 and cKO–CCI = 15.

*Elevated plus maze.* The elevated plus maze consists of two open arms (30 × 5 cm) and two closed arms (30 × 5 × 15 cm) extending from a central area (5 × 5 cm). Photo beams embedded at arm entrances register movements. Room illumination was ~5 lux. The test began by placing the subject mouse in the center, facing a closed arm. The mouse was allowed to freely explore the maze for 5 min. Time spent in the open arms and closed arms, the junction, and the number of entries into the open arms and closed arms, were automatically scored by the MED-PC V 64 bit Software (Med Associates). At the end of the test, the mouse was gently removed from the maze and returned to its home cage. The maze was cleaned with 70% ethanol and wiped dry between subjects.

*Open-field test.* The open field is the most commonly used test for spontaneous exploratory activity in a novel environment, incorporating measurements of locomotion and anxiety-like behaviors. The open-field test was performed adopting published protocols[40]. Exploration was monitored during a 60 min session with Activity Monitor Version 7 tracking software (Med Associates Inc.). Briefly, each mouse was gently placed in the center of a clear Plexiglas arena (27.31 × 27.31 × 20.32 cm, Med Associates ENV-510) lit with dim light (~5 lux), and was allowed to ambulate freely. Infrared (IR) beams embedded along the x, y, z axes of the arena automatically track distance moved, horizontal movement, vertical movement, stereotypies, and time spent in the center zone. Data were analyzed in six, 5-min time bins. Areas were cleaned with 70% ethanol and thoroughly dried between trials.

*Morris water maze acquisition and reversal.* Spatial learning and memory were assessed in the Morris water maze[40]. No visible trials were run before or after hidden platform trials in this study. The 122-cm circular pool was filled 45 cm deep with tap water and rendered opaque with the addition of nontoxic white paint (Crayola). Water temperature was maintained at 23 °C ± 1. The proximal cue was one sticker taped on the inner surface of the pool, ~20 cm above the water surface. Trials were videotaped and scored with Ethovision XT 12 (Noldus). Acquisition training consisted of four trials a day for seven days. Each training trial began by lowering the mouse into the water close to the pool edge, in a quadrant that was either right of, left of, or opposite to, the target quadrant containing the platform (12 cm in diameter). The start location for each trial was alternated in a semi-random order for each mouse, using a random number generator and to avoid dropping the subject in the quadrant that contained the hidden platform. The hidden platform remained in the same quadrant for all trials during acquisition training for a given mouse but varied across subject mice. Mice were allowed a maximum of 60 s to reach the platform. A mouse that failed to reach the platform in 60 s was guided to the platform by the experimenter. Mice were left on the platform for ~15 s before being removed. After each trial, the subject was placed in a cage lined with absorbent paper towels and allowed to rest under an infrared heating lamp for 1 min. Two hours after the completion of the last training trial, the platform was removed and mice were tested in a 60 s probe trial. Parameters recorded during training days were latency to reach the platform, total distance traveled, and swim speed. Time spent in each quadrant and the number of crossings over the trained platform location and over analogous locations in the other quadrants were used to analyze probe trial performance. All trials were recorded and analyzed with the Ethovision XT video tracking software (Noldus Information Technology Inc).

*Fear conditioning.* Fear conditioning was assessed following the published protocols[40]. Training and conditioning tests are conducted in two identical chambers (Med Associates, E. Fairfield, VT) that were calibrated to deliver identical foot shocks. Each chamber was 30 × 24 × 21 cm with a clear polycarbonate front wall, two stainless sidewalls, and a white opaque back wall. The bottom of the chamber consisted of a removable grid floor with a waste pan underneath. When placed in the chamber, the grid floor connected with a circuit board for delivery of the scrambled electric shock. Each conditioning chamber was placed inside a sound-attenuating environmental chamber (Med Associates). A camera mounted on the front door of the environmental chamber recorded test sessions which were later scored automatically, using the VideoFreeze software (Med Associates, E.

Fairfield, VT). For the training session, each chamber was illuminated with a white house light. An olfactory cue was added by dabbing a drop of imitation lemon flavoring solution on the metal tray beneath the grid floor. The mouse is placed in the test chamber and allowed to explore freely for 2 min. A pure tone (5 kHz, 80 dB) which serves as the conditioned stimulus (CS) was played for 30 s. During the last 2 s of the tone, a foot shock (0.5 mA) was delivered as the unconditioned stimulus (US). Each mouse received three CS-US pairings, separated by 90 s intervals. After the last CS-US pairing, the mouse was left in the chamber for another 120 s, during which freezing behavior is scored by the VideoFreeze software. The mouse was then returned to its home cage. Contextual conditioning is tested 24 h later in the same chamber, with the same illumination and olfactory cue present but without foot shock. Each mouse was placed in the chamber for 5 min, in the absence of CS and US, during which freezing is scored. The mouse was then returned to its home cage. Cued conditioning is conducted 48 h after training. Contextual cues were altered by covering the grid floor with a smooth white plastic sheet, inserting a piece of black plastic sheet bent to form a vaulted ceiling, using near-infrared light instead of white light, and dabbing vanilla instead of lemon odor on the floor. The session consisted of a 3 min free exploration period followed by 3 min of the identical CS tone (5 kHz, 80 dB). Freezing was scored during both 3 min segments. The mouse was then returned to its home cage. The chamber was thoroughly cleaned of odors between sessions. % freezing on day 1 was analyzed to indicate the immediate reaction to receiving foot shocks, % freezing on day 2 and day 3 was analyzed to reflect contextual conditioning and cued conditioning, respectively.

**Statistics and reproducibility.** All statistical analyses excluding behavioral tests were performed using Prism (ver. 9, GraphPad). Results are presented as mean ± SEM. The normality of the result was analyzed using the Shapiro–Wilk test before running appropriate statistical tests. Two-group comparisons were analyzed by the unpaired two-tailed $t$ test or nonparametric Mann–Whitney test. The data from Sholl analyses were analyzed using two-way ANOVA (SPSS Statistics, IBM). A value of $P$ less than 0.05 was considered statistically significant.

For behavioral results, statistical analyses and graphs were generated in SigmaPlot (Systat Software, Inc.). WT and cKO were analyzed separately. A two-way RM ANOVA, with time and operation (sham vs. CCI) as the two factors, was performed on distance traveled, center time, and vertical movement data collected from the open-field test. For Morris Water Maze data, latency to platform, swim distance, and swim speed were all analyzed using two-way RM ANOVA with the day as the within-subject factor and operation (sham vs. CCI) as the between-subject factor. Mean % time spent in each quadrant for each group during the probe trial was analyzed using One-way RM ANOVA, followed by Dunnett's post hoc test, to see if the animals spent significantly more time in the trained quadrant than in the other three quadrants while platform crossings during the probe trials were analyzed using an unpaired $t$ test. The percentage of time spent freezing during the contextual and cued phases of fear conditioning was analyzed using two-way ANOVA.

To reduce the experimental bias and enhance the reproducibility, animal behavioral tests were performed by individuals blind to the treatments and genotypes. The animal behaviors were recorded digitally and scored using automated systems (Med Associates and Noldus).

**Reporting summary.** Further information on research design is available in the Nature Research Reporting Summary linked to this article.

## Data availability
The supporting data in the study are available in Supplementary Data 1. The original DNA gel picture is provided in Supplementary Fig. 1. All other data are available from the corresponding authors upon reasonable request.

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

## Acknowledgements

This research was supported by the National Institutes of Health/National Institute of Neurological Disorders and Stroke Grants R56-NS-089523 (S.G.K.) and R01-NS-095803 (S.G.K.) and the Paul Allen Foundation (S.G.K.).

## Author contributions

T.S.Y. and S.G.K. contributed to experimental design, T.S.Y., Y.T., and E.P.S. contributed in surgery and neuronal morphology analysis, T.S.Y. and S.C. contributed in characterizing generated conditional knockout mice, T.S.Y., E.E.R., and M.Y. contributed in animal behavioral tests and analyses, T.S.Y., M.Y., and S.G.K. contributed in writing manuscript.

## Competing interests

The authors declare no competing interests.
