## [Peer Review File · Communications Biology]

Reviewers' comments:

Reviewer #1 (Remarks to the Author):

This manuscript deals with potential roles of APOE in neuronal dendritic formation in newly born neurons after TBI. The authors found that postnatal ablation of astrocytic APOE reduces the dendritic complexity of newborn dentate gyrus neurons after TBI. Along with this anatomical defects, APOE conditionally deficient mice after injury showed additive defects in spatial memory tests. As the author noted, the use of a conditional deletion of APOE only in astrocytes in a temporally controlled manner bypasses several caveats in previous studies, potentially providing clear evidence for roles of astrocyte-derived APOE in neuronal process formation after injury. However, there are several major points that reduce the enthusiasm, which need to be addressed to strengthen the author's claims.

Major points

1. APOE KO efficiency needs to be validated with additional methods other than IHC. In addition, it would be interesting to perform western analysis to confirm how much portion of APOE protein gets reduced from the brain by deleting astrocytic APOE.

2. Since Golgi-Cox staining labels random neurons, the data generated from Golgi-Cox staining would represent the combined effects of both newborn and pre-existing neurons. This point is not clearly presented. Moreover, it should be presented how many new neurons can be born during normal and injury states and whether this can affect the data generated from Golgi-Cox staining.

3. Unlike sham-control, distribution of dendrites with Golgi-Cox staining was not changed in APOE cKO mice after injury. The authors need to explain this difference.

4. The dendritic complexity of newborn neurons does not appear to show the strong co-relation with behavior results, since complexity of dendritic trees from APOE deleted newborn neurons without TBI was slightly increased even though cKO animals without TBI showed defects in spatial memory. Therefore, another features of DG neurons, such as spine density should be examined.

Reviewers #2-3 (Remarks to the Author):

This study by Yu et al., assesses the impact of astrocytic ApoE on hippocampal neurogenesis and behavior after traumatic brain injury. To overcome limitations of studies using full ApoE knockout or humanized mice, the authors generated a new mouse line with loxP sites flanking exon 1 and 2 of ApoE, enabling cell-type specific and timed ApoE deletion. The morphology of dendritic neurons in the dentate gyrus of the hippocampus was assessed and several behavioral assays including Morris Water Maze, open field, elevated plus maze, and fear conditioning were used. Overall, data were obtained rigorously but adjustments to data presentation are needed, and some statistical analysis might have to be reassessed. There are questions with regard to the chosen experimental design. Further, the study is largely observational without any mechanistic insight as to how ApoE contributes to reduced complexity of the dendritic tree or how modified dendritic arborization influences behavior given the myriad other mechanisms induced by TBI.

Major points:

- Conceptually, it is unclear how deleting ApoE relates to the human ApoE isoforms which either confer protection or risk for the development of neurodegenerative disease. In the discussion the authors argue that the ApoE4 human isoform acts as dominant negative but do not provide references. To support a statement regarding findings in humanized ApoE4 mice reference 27 is used, which does not use humanized ApoE4 mice. The authors further infer that ApoE deletion phenocopies humanized ApoE4 mice but in the same paragraph point out that driving overexpression of ApoE4 in astrocytes does not phenocopy their findings. This comes across as contradictory and no further explanation as to these differences is provided.
- It is unclear why ApoE is depleted at 3 weeks of age during postnatal development instead of in adult animals when all potential developmental effects can be avoided. A similar question pertains to the age at which the TBI was conducted (6 weeks of age). Mice do not reach maturity until 8-12

weeks of age and biomechanics of TBI differ significantly dependent on age. The authors should state their rationale for this experimental design.

- It should be clarified what the evidence is that dendritic morphology after injury is responsible for performance in the chosen behavioral tasks. The authors present two datasets demonstrating that ApoE influences dendritic morphology and hippocampal dependent learning. However, there is not even correlative data presented that the two are connected because histology and behavior were performed in different cohorts. Was histology assessed in animals after behavioral assays were completed? The study is exclusively descriptive, there is no mechanisms linking ApoE loss, subtle differences in arborization of newborn neurons and behavioral changes.
- There is no block/mimic data suggesting that the morphological changes in newborn neurons are causally related to the changes in behavior. This might be beyond the scope of this study but this distinction should be made clear in the manuscript.

Minor points:

- For the behavioral assays male and female mice were used but it is unclear if both sexes were used for the other experiments.
- It should be clarified how the starting location in the Morris Water Maze was randomized. Currently the text states "semi-randomized" but no definition of the process is given.
- Exact p-values should be included consistently.
- Was normality tested to determine which statistical test?
- Instead of bar graphs individual data points should be represented in scatter plots.
- In Fig. 2j the y-axis it is not sufficiently descriptive. It should be made clear that the graph refers to ApoE-positive cells.
- In Fig.3 panels B, D, F, H are hard to make out, especially in a printed version but even on a computer after magnification they are almost impossible to see.
- In the text, the nomenclature for the mouse is inconsistent.
- In the methods, micrometer is sometimes labeled "mm" or "um".
- There also are several other typos throughout the text.
- For the Morris Water Maze, the reader is directed to previous studies cited but no references are included.
- In the fear conditioning assay, as per the methods section an unpaired t-test was used but there are more than two groups compared in Figure 7D. Fig. 7D is hard to read without refereeing back and forth between the text and the figure. The figure panel should be labeled better.

Reviewer #4 (Remarks to the Author):

The manuscript addresses the role of ApoE on the effects of TBI on neurons in the hippocampus and associated cognitive tasks. In summary, the authors found that the combination of ApoE removal and TBI resulted in the most severe outcomes in terms of memory tasks and dendritic architecture in the dentate gyrus.

This is a well written manuscript and the experiments describe well controlled tests of the intersection of ApoE and traumatic brain injury in an important brain region for neurodegeneration. Although some of the results may not have been predictable, overall the importance of ApoE, in particular with post-TBI neurogenesis, was elucidated. Basically, an elegant cause and effect test of ApoE involvement was developed with a conditional cre-lox system. They confirmed that the conditional system worked as expected by IHC for ApoE in injured and sham control mice.

1. In Figures 3 and 4, it would be useful to display some direct comparisons between the sham and injured for the Sholl analysis curves. Additional literature could be included noting the injury induced effects on dendritic changes in the dentate gyrus.

2. In Figure 2 panels, B, D, F, and H and in Figure 3, panels A-D, the dendrites are very difficult to see and the illustrations would benefit from different channel color choices or inversion.

Minor typographical items:

3. Font discrepancies resulted in um displaying as mm in several places in the materials and

methods.

4. 122cm should read 122 cm

5. Prizm should read Prism

6. In the Figure 6 legend, should "E, F, F, G" read "B, D, F, H"?

Reviewers' comments:

Reviewer #1 (Remarks to the Author):

This manuscript deals with potential roles of APOE in neuronal dendritic formation in newly born neurons after TBI. The authors found that postnatal ablation of astrocytic APOE reduces the dendritic complexity of newborn dentate gyrus neurons after TBI. Along with this anatomical defects, APOE conditionally deficient mice after injury showed additive defects in spatial memory tests. As the author noted, the use of a conditional deletion of APOE only in astrocytes in a temporally controlled manner bypasses several caveats in previous studies, potentially providing clear evidence for roles of astrocyte-derived APOE in neuronal process formation after injury. However, there are several major points that reduce the enthusiasm, which need to be addressed to strengthen the author's claims.

Major points

1. APOE KO efficiency needs to be validate with additional methods other than IHC. In addition, it would be interesting to perform western analysis to confirm how much portion of APOE protein gets reduced from the brain by deleting astrocytic APOE.

We agree that in general IHC is non-quantitative and that in order to quantify how efficient our ApoE knockdown was, we have now employed quantitative RT-PCR. We chose this technique over western analysis because of the relatively small amounts of protein extracted from individual mouse hippocampus. We have added these results as new data in Figure 2A and demonstrate that tamoxifen-dependent ApoE mRNA expression was reduced by approximately 80% in floxed ApoE animals exposed to tamoxifen compared to those exposed to vehicle only.

2. Since Golgi-Cox staining labels random neurons, the data generated from Golgi-Cox staining would represent the combined effects of both newborn and pre-existing neurons. This point is not clearly presented. Moreover, it should be presented how

many new neurons can be born during normal and injury states and whether this can affect the data generated from Golgi-Cox staining.

We appreciate how relevant this is to the overall story and we have added an additional description on page 18 lines 17-22 to elucidate this point in the context of how overall Golgi-Cox stained neurons correlate with the number of newly generated ones. As we inferred in the original manuscript and now clearly demonstrate, the vast majority of Golgi-Cox stain neurons are pre-existing though we cannot exclude the possibility of having a small fraction of newly born neurons labelled as well.

3. Unlike sham-control, distribution of dendrites with Golgi-Cox staining was not changed in APOE cKO mice after injury. The authors need to explain this difference.

This is a critical point and we appreciate the opportunity to more clearly elucidate. We have performed further analysis that uses surgery as the main factor to evaluate the data using a Two-way ANOVA (Supplementary Figure 1) comparing control animals with ApoE conditional knockout ones. Here we demonstrate that CCI itself plays a critical role in causing impairments in dendritic trees of the Golgi-Cox stained pre-existing neurons whether or not ApoE is present. We have added these additional data to Supplementary Figure 1 and have elaborated upon this finding in the Results section, page 19 lines 15-24.

4. The dendritic complexity of newborn neurons does not appear to show the strong correlation with behavior results, since complexity of dendritic trees from APOE deleted newborn neurons without TBI was slightly increased even though cKO animals without TBI showed defects in spatial memory. Therefore, another features of DG neurons, such as spine density should be examined.

We agree with this point and have performed further analysis of spine density in 4 separate groups (new Figure 5). Here we demonstrate that astrocytic ApoE plays a clear role in spine density in both normal development of adult-born neurons and in the

context of traumatic injury. In addition to this new Figure 5 demonstrating an ApoE-dependent effect, we have also elaborated upon it in the context of previous work and its relevance to the present study (pages 20, lines 20-25 and 21, lines 1-13).

Reviewers #2-3 (Remarks to the Author):

This study by Yu et al., assesses the impact of astrocytic ApoE on hippocampal neurogenesis and behavior after traumatic brain injury. To overcome limitations of studies using full ApoE knockout or humanized mice, the authors generated a new mouse line with loxP sites flanking exon 1 and 2 of ApoE, enabling cell-type specific and timed ApoE deletion. The morphology of dendritic neurons in the dentate gyrus of the hippocampus was assessed and several behavioral assays including Morris Water Maze, open field, elevated plus maze, and fear conditioning were used. Overall, data were obtained rigorously but adjustments to data presentation are needed, and some statistical analysis might have to be reassessed. There are questions with regard to the chosen experimental design. Further, the study is largely observational without any mechanistic insight as to how ApoE contributes to reduced complexity of the dendritic tree or how modified dendritic arborization influences behavior given the myriad other mechanisms induced by TBI.

Major points:

- Conceptually, it is unclear how deleting ApoE relates to the human ApoE isoforms which either confer protection or risk for the development of neurodegenerative disease. In the discussion the authors argue that the ApoE4 human isoform acts as dominant negative but do not provide references. To support a statement regarding findings in humanized ApoE4 mice reference 27 is used, which does not use humanized ApoE4 mice. The authors further infer that ApoE deletion phenocopies humanized ApoE4 mice but in the same paragraph point out that driving overexpression of ApoE4 in astrocytes does not phenocopy their findings. This comes across as contradictory and no further explanation as to these differences is provided.

We apologize for the confusion. The discussion on page 25 and cited references there were meant to emphasize the caveats of the lack of specific means to study the role of ApoE in the brain. The discussion regarding the ApoE deletion phenocopies humanized ApoE4 mice should be restricted to what is known in the context of adult hippocampal neurogenesis. We have replaced this confusing paragraph from the original version and now emphasize the observed differences in neurogenesis due to ApoE-deficient and humanized ApoE4 and how this current work is consistent with previous studies demonstrating that ApoE-deficiency and the presence of human ApoE4 largely phenocopy each other in the context of hippocampal neurogenesis (page 26, lines 18-25 and page 27, lines 1-10).

- It is unclear why ApoE is depleted at 3 weeks of age during postnatal development instead of in adult animals when all potential developmental effects can be avoided. A similar question pertains to the age at which the TBI was conducted (6 weeks of age). Mice do not reach maturity until 8-12 weeks of age and biomechanics of TBI differ significantly dependent on age. The authors should state their rationale for this experimental design.

The majority of neurogenesis and gliogenesis is complete by 3 weeks of age in murine models, though myelination remains incomplete. We chose this timepoint to administer tamoxifen in order to get past most of the key elements of brain development, but at a time when hippocampal neurogenesis is the primary mode of neurogenesis occurring. We performed injuries at 6 weeks of age when hippocampal neurogenesis is particularly robust as we have been performing these kinds of studies for the past 20 years and have intentionally chosen timepoints where we believe we would see the most robust neurogenic phenotype. Although we agree with the reviewer that TBI biomechanics differ with age, it does not make studying younger animals less important. As a physician-scientist formally trained as a pediatrician, my lab and I have always sought to study the impact of TBI modeling on more juvenile models, which are more relevant to the more dynamic and relatively less studied pediatric population. We have added

further description around this justification in the results section in page 16 (lines 16-19, 21-22).

- It should be clarified what the evidence is that dendritic morphology after injury is responsible for performance in the chosen behavioral tasks. The authors present two datasets demonstrating that ApoE influences dendritic morphology and hippocampal dependent learning. However, there is not even correlative data presented that the two are connected because histology and behavior were performed in different cohorts. Was histology assessed in animals after behavioral assays were completed? The study is exclusively descriptive, there is no mechanisms linking ApoE loss, subtle differences in arborization of newborn neurons and behavioral changes.

We appreciate these comments and have tried to clarify our observations. We also have added additional data demonstrating that conditional depletion of ApoE affects not just dendritic arborization, but also spine density on newly born neurons (new Figure 5). Together, this may help explain the observed behavioral deficits though in itself does not provide underlying mechanisms about why this is occurring in the context of conditional ApoE deletion. We have enhanced the discussion in (page 26, lines 21-25, page 27, lines 1-10) to emphasize that the observed impairments in Morris water maze reversal was due to the lack of astrocytic ApoE, which correlates to both reduced spine density and impaired dendritic arborizations.

- There is no block/mimic data suggesting that the morphological changes in newborn neurons are causally related to the changes in behavior. This might be beyond the scope of this study but this distinction should be made clear in the manuscript.

We agree with the reviewer that our observed morphological changes in dendritic complexity (and the newly added spine density) would best be demonstrated to lead to the observed behavioral deficits if we could reverse these changes and show corresponding reversals in behavior. This is indeed beyond the scope of the study here though we have now added new discussion related to how best to determine the

mechanistic links of structure to behavior (page 28, lines 18-25).

Minor points:

- *For the behavioral assays male and female mice were used but it is unclear if both sexes were used for the other experiments.*

Yes, we added this point (page 7, line 7).

- *It should be clarified how the starting location in the Morris Water Maze was randomized. Currently the text states “semi-randomized” but no definition of the process is given.*

The sentence on page 12 (lines 15-17) has been modified to read “using a random number generator and to avoid dropping the subject in the quadrant that contained the hidden platform.”

- *Exact p-values should be included consistently.*

This has been rectified within the Results section for all statistical analyses.

- *Was normality tested to determine which statistical test?*

Yes, and we emphasized that in the section “Statistical analysis.”

- *Instead of bar graphs individual data points should be represented in scatter plots.*

All the graphs have been replaced with scatter plots.

- *In Fig. 2j the y-axis it is not sufficiently descriptive. It should be made clear that the graph refers to ApoE-positive cells.*

The figure has been revised accordingly.

- In Fig.3 panels B, D, F, H are hard to make out, especially in a printed version but even on a computer after magnification they are almost impossible to see.

The figure has been revised accordingly.

- In the text, the nomenclature for the mouse is inconsistent.

The nomenclature was has been uniformly changed for consistency.

- In the methods, micrometer is sometimes labeled “mm” or “um”.

This has been corrected.

- There also are several other typos throughout the text.

These have been corrected.

- For the Morris Water Maze, the reader is directed to previous studies cited but no references are included.

The reference was added.

- In the fear conditioning assay, as per the methods section an unpaired t-test was used but there are more than two groups compared in Figure 7D. Fig. 7D is hard to read without refereeing back and forth between the text and the figure. The figure panel should be labeled better.

We apologize for the confusion. The text on page 15 (lines 4-5) has been modified to read “Percentage of time spent freezing during the contextual and cued phases of Fear Conditioning was analyzed using two-way ANOVA.” The text on page 23 (line 14) has been revised, to read “with CCI mice showing less freezing than animals without sham operation”. Figure 8 has been modified to indicate the operation effect on contextual conditioning on Day 2.

Reviewer #4 (Remarks to the Author):

The manuscript addresses the role of ApoE on the effects of TBI on neurons in the hippocampus and associated cognitive tasks. In summary, the authors found that the combination of ApoE removal and TBI resulted in the most severe outcomes in terms of memory tasks and dendritic architecture in the dentate gyrus.

This is a well written manuscript and the experiments describe well controlled tests of the intersection of ApoE and traumatic brain injury in an important brain region for neurodegeneration. Although some of the results may not have been predictable, overall the importance of ApoE, in particular with post-TBI neurogenesis, was elucidated. Basically, an elegant cause and effect test of ApoE involvement was developed with a conditional cre-lox system. They confirmed that the conditional system worked as expected by IHC for ApoE in injured and sham control mice.

1. In Figures 3 and 4, it would be useful to display some direct comparisons between the sham and injured for the Sholl analysis curves. Additional literature could be included noting the injury induced effects on dendritic changes in the dentate gyrus.

We appreciate the observation and we have added these suggested analyses in Supplementary Figures 1 and 2. These results are now described on pages 19 (lines 15-24) and 21 (lines 3-13).

2. In Figure 2 panels, B, D, F, and H and in Figure 3, panels A-D, the dendrites are very difficult to see and the illustrations would benefit from different channel color choices or inversion.

Thank you for the comments, and we have made the background white for better visualization of the analyzed dendritic trees as shown in Figure 3 B, D, E, and H and in Figure 4 A-D.

Minor typographical items:

3. Font discrepancies resulted in *um* displaying as *mm* in several places in the materials and methods. They were corrected.
4. 122cm should read 122 cm It has been corrected.
5. Prizm should read Prism It has been corrected.
6. In the Figure 6 legend, should “E, F, F, G” read “B, D, F, H”? It has been corrected.

Reviewers' comments:

Reviewer #1 (Remarks to the Author):

The authors have addressed most of previous concerns.

Reviewers #2-3 (Remarks to the Author):

By and large the authors addressed our comments. However, we remain concerned about the statistics used throughout the study and the legibility of the figures. Many axes labels are not legible. The authors now include which statistical test was run for each data set, which makes it even more obvious that tests are not always appropriately used. For example (this is not a complete list):

Fig. 2b uses an unpaired t-test but 4 groups are plotted in the graph. These four groups should be compared statistically with genotype and injury as independent factors.

Fig. 3 J, M should be compared in one statistical analysis and be plotted in the same graph, the same goes for Fig. 3 K, N

New Fig. 5 should compare the 4 groups in one statistical analysis and plot the data in the same graph.

Fig. 3 and 4, Sholl analysis uses a 2-way ANOVA with radius as one of the factors, but this is a continuous variable.

Other points:

The abstract and significance statement states that "ApoE is required for functional injury-induced neurogenesis" but the authors assess the morphology of new-born neurons instead of the genesis of these neurons; we suggest specifying because one would otherwise expect assessment of numbers of newborn neurons, proliferative capacity of progenitors etc.

In the significance statement the wording "via unknown mechanisms" should be removed to not suggest that this manuscript reveals such mechanisms.

Page 5, line 13, the authors state that "Most remarkably, these morphologic changes appear to be functionally relevant, as demonstrated by the pronounced spatial memory deficits in ApoE conditionally deficient mice with CCI, in both the acquisition and reversal phases of the Morris water maze test." Yet to address our concern about these two data sets simply co-occurring the authors specified in the discussion "Whether the poor performance in reversal Morris water maze is correlated with impaired newborn neurons remains unclear." The introduction needs to be adjusted.

In the results section the authors state that they aim to bypass developmental effects but also state that ApoE KO mimic the human ApoE4 allele, which would be present during development in humans. Perhaps, the main point here is the cell-type specific deletion of ApoE?

The authors administer Tamoxifen at 3 weeks of age to Aldh1l1-CreERT2 mice. In reference 8, Tamoxifen was administered to adult mice. The study specifically states that controls are needed for different ages/ time points. Since a reporter was used, can the authors make explicit that they didn't see recombination in newborn neurons in the dentate gyrus?

Reviewers' comments:

Reviewer #1 (Remarks to the Author):

The authors have addressed most of previous concerns.

Reviewers #2-3 (Remarks to the Author):

By and large the authors addressed our comments. However, we remain concerned about the statistics used throughout the study and the legibility of the figures. Many axes labels are not legible. The authors now include which statistical test was run for each data set, which makes it even more obvious that tests are not always appropriately used. For example (this is not a complete list):

Fig. 2b uses an unpaired t-test but 4 groups are plotted in the graph. These four groups should be compared statistically with genotype and injury as independent factors.

Fig.3 J, M should be compared in one statistical analysis and be plotted in the same graph, the same goes for Fig. 3 K, N

New Fig. 5 should compare the 4 groups in one statistical analysis and plot the data in the same graph.

Fig. 3 and 4, Sholl analysis uses a 2-way ANOVA with radius as one of the factors, but this is a continuous variable.

We appreciate the concerns around the statistical analysis. We initially performed these analyses with Prism (GraphPad) software and following consultation with both colleagues who perform more sophisticated analyses than we typically use and a Columbia biostatistician, we reworked the statistics for Figures 2b, 3, 4, and 5 as suggested above using IBM SPSS. While the overall story has not changed, the analysis is clearly now more appropriate to the comparisons we were making. The changes in the document are all noted in blue both in the results and figure legends. We have also increased the size of fonts to improve the legibility.

Other points:

The abstract and significance statement states that “ApoE is required for functional injury-induced neurogenesis” but the authors assess the morphology of new-born neurons instead of the genesis of these neurons; we suggest specifying because one would otherwise expect assessment of numbers of newborn neurons, proliferative capacity of progenitors etc.

We rewrote the sentence to emphasize the role of astrocytic ApoE in newborn neurons for proper spine density instead of the general term “neurogenesis”. See lines 3-7 page 3.

In the significance statement the wording “via unknown mechanisms” should be removed to not suggest that this manuscript reveals such mechanisms.

The statement was removed accordingly, lines 3-7 page 3.

Page 5, line 13, the authors state that “Most remarkably, these morphologic changes appear to be functionally relevant, as demonstrated by the pronounced spatial memory deficits in ApoE conditionally deficient mice with CCI, in both the acquisition and reversal phases of the Morris water maze test.” Yet to address our concern about these two data sets simply co-occurring the authors specified in the discussion “Whether the poor performance in reversal Morris water

maze is correlated with impaired newborn neurons remains unclear.” The introduction needs to be adjusted.

We have re-written this section, page 5, lines 9-17.

In the results section the authors state that they aim to bypass developmental effects but also state that ApoE KO mimic the human ApoE4 allele, which would be present during development in humans. Perhaps, the main point here is the cell-type specific deletion of ApoE?

We have re-written this section to reflect the above clarification. See lines 2-3, page 16.

The authors administer Tamoxifen at 3 weeks of age to Aldh111-CreERT2 mice. In reference 8, Tamoxifen was administered to adult mice. The study specifically states that controls are needed for different ages/ time points. Since a reporter was used, can the authors make explicit that they didn't see recombination in newborn neurons in the dentate gyrus?

We have clarified this issue in the Discussion where we highlight the limitations of our approach using Aldh111-creERT mice. See pages 23-24, lines 21-25 and 1-4.